# Genetic Effects Analysis of QTLs for Rice Grain Size Based on CSSL-Z403 and Its Dissected Single and Dual-Segment Substitution Lines

**DOI:** 10.3390/ijms241512013

**Published:** 2023-07-27

**Authors:** Guangyi Xu, Keli Deng, Jinjin Yu, Qiaolong Li, Lu Li, Aoni Xiang, Yinghua Ling, Changwei Zhang, Fangming Zhao

**Affiliations:** Rice Research Institute, Academy of Agricultural Science, Engineering Research Center of South Upland Agriculture, Ministry of Education, Southwest University, Chongqing 400715, China; 15520096684@163.com (G.X.);

**Keywords:** rice, QTL, chromosomal segment substitution line, grain size, additive and epistatic effect

## Abstract

Rice chromosomal segment substitution lines (CSSLs) are ideal materials for studying quantitative traits such as grain size. Here, a rice large-grain CSSL-Z403 was identified among progeny of the recipient Xihui18 and the donor Jinhui35 based on molecular marker-assisted selection. Z403 carried 10 substitution segments with average length of 3.01 Mb. Then, a secondary F_2_ population derived from a cross between Xihui18 and Z403 was used to map quantitative trait loci (QTL) for grain size. Six QTLs distributed on chromosomes 5, 6, 7, 9 and 12 were detected. Finally four single-segment substitution lines (SSSLs) and two dual-segment substitution lines (DSSLs) carrying these target QTLs were constructed, and 10 novel QTLs were identified by four SSSLs. The large grain of Z403 was controlled at least by *qGWT5*, *qGWT7*, *qGWT9* and *qGWT12*, and its grain weight was influenced through grain length QTL such as *qGL5*, *qGL6*, *qGL9* and *qGL12*, as well as grain width QTL such as *qGW5*, *qGW7*, *qGW9* and *qGW12*. Among 16 QTLs, four QTLs including *qGL6*, etc., might be novel compared with the reported documents. Again, positive or less negative epistatic effects between two non-allelic QTLs (additive effect > 0) may assist screening the genotype with larger grain size in further selection.

## 1. Introduction

Rice is a staple food crop all over the world. The targets of rice improvement have become the combined targets of high yield and good quality [1]. Grain weight, number of grains per panicle, and number of effective panicles per plant are the main factors affecting rice yield [2]. But the grain weight, usually used to evaluate grain size, may be correlated with several characters including grain length, grain width, and grain thickness [3]. Again, rice grain size is also closely related to appearance quality. Thus, mapping and cloning of these quantitative trait loci (QTLs) for grain size are of great significance for high-yield and good-quality breeding in rice.

Unlike quality traits, the inheritance mechanisms of grain size are complex. They are controlled by many quantitative trait loci (QTLs) with minor effects and are easily affected by environment factors [4]. In order to improve the accuracy of QTL mapping and achieve the integration of QTL in breeding by design, secondary mapping populations such as CSSL and near-isogenic lines (NIL) have gradually attracted researchers– attention. A chromosomal segment substitution line (CSSL) is a type of backcross that self-crosses to generate a near-isogenic line with one or several specific marker-defined homozygous chromosome segments from a donor parent. They are a complete library of introgression lines with chromosomal segments usually of a distant genotype in an adapted background and are valuable genetic resources for basic and applied research on improvement of complex traits [5,6,7,8,9,10]. Thus, rice CSSLs are ideal materials for creating natural occurring variations and genetic dissection and pyramids of QTLs for traits such as rice yield and quality, which provides a bridge for connection of different subjects such as quantitative genetics, Mendelian genetics, molecular genetics, and molecular breeding.

Several advantages of the use of CSSLs have been examined in terms of genetic analysis, molecular cloning of QTLs and marker-assisted selection in rice breeding [11]. To date, some QTLs for grain size have been identified by CSSLs or SSSLs in rice. Eleven QTLs for grain size distributed across chromosomes 3, 7 and 8 were identified by a CSSL-Z563 with seven substitution segments from donor parent Huhan3 based on the genetic background of Xihui18, and then were dissected into four single-segment substitution lines (SSSLs), of which a major QTL *qGL3-2* was fine-mapped into a 696 kb region of chromosome 3 containing five candidate genes [6]. Sixteen QTLs for rice grain size distributed on chromosomes 1, 2, 3, 5 and 12 were detected by a CSSL-Z431 carrying six-segment substitution from donor parent Huhan3 in the genetic background of Xihui18, and then were dissected into five SSSLs, and finally *qGL3* and *qGW5* were fine-mapped and predicted the candidate genes [12]. Nine QTLs for grain size and chalky degree (CD) distributed on chromosomes 3, 5 and 11 were identified by a short-wide grain CSSL-Z414 with four substitution segments from Huhan3 in the genetic background of Xihui18, and then were dissected into six SSSLs, of which *qGL11* was fine-mapped [13]. *qLG9* for seed longevity was fine-mapped in a 30 kb interval of chromosome 9 by a rice near-isogenic line (NIL) of the Nipponbare genome sequence [14]. Three major QTLs (*qGP4*, *qGP5*, and *qBL4*) conferring seed growth capacity were detected by using a CSSL constructed from *japonica* rice 9311 as the recipient parent and *indica* rice Nanyangzhan as the donor parent [15]. Seven QTLs for rice grain length and width distributed across chromosomes 2, 3, 5, 8 and 10 were identified by 33 CSSLs from donor *Oryza rufipogon (W0106)* in the genetic background of *japonica* cultivar Koshihikari [16]. Twenty-two QTLs affecting rice grain size were detected to be distributed on eight chromosomes except chromosomes 6, 9, 11 and 12 by SSSLs from Guanglu’ai 4 in the genetic background of Nipponbare [17]. Sixty-four QTLs associated with increases in grain length, grain width, grain thickness, 1000-grain weight, etc., were identified by 34 chromosome segment substitution lines of *O. glaberrima* in the background of the elite *japonica* cultivar Koshihikari [18]. A QTL for grain weight (*qPW11.1*) distributed on chromosome 11 was detected by a set of CSSLs from *O. nivara* IRGC81832 in the genetic background of *Oryza sativa* Swarna [19]. In addition, a CSSL library of a U.S. weedy rice accession “PSRR-1” was developed with genome-wide coverage in an adapted rice cultivar “Bengal” background, and several CSSLs with desirable agronomic traits (e.g., longer grains, higher seed weights, etc.) were found [20]. Again, 40 chromosome segment substitution lines (CSSL) of *O. barthii* were developed in the background of the elite *japonica* cultivar Koshihikari, and candidate chromosome segments that harbor QTLs associated with grain size-related traits were identified [21]. Four introgression lines were identified from *Oryza rufipogon* based on the genetic background of IR64, which showed higher yield due to an increase in grain weight [22]. Thus, these QTLs for grain size-related traits dissected into SSSLs to some degree lay a good foundation for gene functional analysis of grain size in molecular genetics and then application in breeding by design. Zhang et al. has made considerable achievements in rice breeding by design using the HJX74-SSSL platform consisting of 2360 SSSLs and argued that target chromosome-segment substitution is a way to breed by design [23]. 

Apparently, grain size is controlled by many genes, and different favorable alleles are scattered in various germplasm resources of different ecological areas. Therefore, it is necessary to construct CSSLs for breeding plans to meet the rice demand in specific ecological districts. Xihui18, an *indica* rice restorer line, was bred by the Rice Research Institute of Southwest University, China. It has many advantages, such as high general combining ability, good flowering habit and long panicles, and multiple grains per panicle. However, its grains are long and narrow. Jinhui35, an *indica* rice restorer line, is characterized by large panicles and wide and large grains. To achieve whole-genome rice breeding by molecular design in the special ecological area of southwestern China, we are constructing a CSSL platform in the genetic background of Xihui18. Here, a rice large grain CSSL-Z403 with 10 substitution segments from Jinhui35 was used as material to clarify the following theoretical issues: Since Z403 contains 10 substitution segments from Jinhui35, how many QTLs will respond to the large grain in Z403, and how are they distributed in these substitution segments? If more than one QTL controls the same trait, will they be independently inherited or interact epistatically, and how to influence the phenotype of these traits for these various non-allelic QTL combinations? Which of these QTLs have been reported, and which are novel, detected in the study? The findings will be important for exploring more beneficial QTLs for grain size and then for breeding by design based on the SSSL platform. 

## 2. Results

### 2.1. Identification of Substitution Segments in Z403

The chromosomal segment substitution line Z403 was developed from *indica* rice restorer line Xihui18 as recipient parent and Jinhui35 as donor parent. The source of the reference genome of Z403 was referred to *indica* sequenced variety “9311”. Based on the previous development of Z403 with whole-genome molecular marker-assisted selection (MAS) of 236 polymorphic single sequence repeats (SSRs), in the paper, 14 polymorphic SSR markers on the substitution segments of Z403 and 70 SSR markers outside the substitution segments were used to detect the substitution segments and assess the homogeneity of genetic background using 10 plants of Z403. The results showed that all of the plants harbored 10 consistent substitution segments, and no residual segments from Jinhui35 were detected. The Z403 plants were indicated to be homozygous. Ten substitution segments of Z403 originating in Jinhui35 were located on rice chromosomes 4, 5, 6, 7, 9, and 12, with a total substitution length of 30.06 Mb. The longest substitution segment was 7.82 Mb, the shortest length was 0.67 Mb, and the average length was 3.01 Mb (Figure 1).

### 2.2. Analysis of Grain Size-Related Traits between Z403 and Xihui18

The plant type (Figure 2a) and panicle type (Figure 2b) of Z403 were similar to those of its recipient Xihui18, which belongs to the large panicle type. The most striking feature of Z403 was that it displayed a larger grain than that of Xihui18 (Figure 2c,d). Compared with Xihui18, the grain width, 1000-grain weight and grain length of Z403 were significantly improved, increasing by 42.38%, 37.80% and 2.65%, respectively (Figure 2e–h). 

### 2.3. Identification of QTLs for Grain Size Using a Secondary F_2_ Population from Xihui18/Z403

Six QTLs harboring grain size were identified, which explained from 8.02% to 19.49% of the phenotypic variation. They were distributed on chromosomes 5, 6, 7, 9 and 12, including *qGL6* and *qGL12* for grain length, *qGW5* for grain width, *qGWT7* and *qGWT12* for 1000-grain weight, and *qRLW9* for ratio of length to width. The grain length of Z403 was mainly controlled by *qGL12* and *qGL6* from Jinhui35, whose additive effects increased the grain length of Z403 by 0.11 mm and 0.08 mm and explained 16.10% and 8.54% of the variation in grain length, respectively. The additive effect of *qGW5* from Jinhui35 increased the grain width of Z403 by 0.09 mm and explained 19.15% of the variation in grain width. The additive effects of *qGWT7* and *qGWT12* from Jinhui35 increased 1000-grain weight of Z403 by 3.99 g and 2.76 g and explained 19.49% and 9.34% of the variation in 1000-grain weight, respectively. The additive effect of *qRLW9* from Jinhui35 increased the ratio of grain length to width by 0.07 and explained 8.02% of the variation in this trait (Table 1).

### 2.4. Verification and Analysis of Additive and Epistatic Effects of QTLs for Grain Size Using the Newly Developed SSSLs and DSSLs

Based on the QTL mapping results, four SSSLs (S1–S4) and two DSSLs (D1–D2) were further developed in the F_3_ population by MAS. S1 carried a substitution segment RM5874--RM2422-RM169--RM60821 on chromosome 5. S2 contained a substituted segment RM5481--RM1135-RM1279--RM7564 on chromosome 7. S3 harbored a substituted segment RM6491--RM3808-RM215--RM2144 on chromosome 9. S4 contained a substituted segment RM247--RM491--RM7119 on chromosome 12. D1 and D2 carried substituted fragments from chromosomes 5 and 7, and chromosomes 9 and 12 of Jinhui35, respectively (Figure 3a).

#### 2.4.1. Verification and Analysis of Additive Effects of QTLs for Grain Size by the SSSLs (S1–S4)

In the four SSSLs, a total of 14 QTLs were detected, of which four QTLs (*qGL12*, *qGW5*, *qRLW9*, *qGWT7*) were verified, indicating their stable inheritance. Only *qGWT12* was not verified by S4, while *qGL6* was not verified because of no corresponding SSSL developed. Thus, 10 novel QTLs (*qGL5*, *qRLW5*, *qGWT5*, *qGW7*, *qRLW7*, *qGL9*, *qGW9*, *qGWT9*, *qGW12*, *qRLW12*) were only detected by four SSSLs. The results indicated that SSSLs with consistent genetic backgrounds had a higher efficiency of QTL detection than the F_2_ population of Xihui18/Z403 (Figure 3a, Table 1).

The grain length (11.12 mm) of S3 with *qGL9* (*a* = 0.23 mm) was significantly longer than that (10.66 mm) of Xihui18, while the grain length (10.50 mm) of S1 carrying *qGL5* (*a* = −0.08 mm) and that (10.28 mm) of S4 containing *qGL12* (*a* = −0.19 mm) were significantly shorter than that (10.66 mm) of Xihui18. S2 without QTL for grain length displayed no significant difference with Xihui18 (Figure 3b). The grain widths (4.29 mm, 3.43 mm, 3.70 mm and 3.31 mm ) of S1 with *qGW5* (*a* = 0.64 mm), S2 carrying *qGW7* (*a* = 0.21 mm), S3 containing *qGW9* (*a* = 0.34 mm) and S4 with *qGW12* (*a* = 0.15 mm) were significantly broader than that (3.01 mm) of Xihui18 (Figure 3c).The ratios of grain length to width (2.45, 3.10, 3.01 and 3.11) of S1 with *qRLW5* (*a* = −0.55), S2 carrying *qRLW7* (*a* = −0.22), S3 containing *qRLW9* (*a* = −0.26) and S4 carrying *qRLW12* (*a* = −0.21) were significantly less than that (3.54) of Xihui18 (Figure 3d). The 1000-grain weights (37.91 g, 29.30 g and 35.14 g) of S1 with *qGWT5* (*a* = 4.65 g), S2 carrying *qGWT7* (*a* = 0.35 g) and S3 containing *qGWT9* (*a* = 3.27 g) were significantly heavier than that (28.61 g) of Xihui18, while the 1000-grain weight (28.79 g) of S4 without QTL for grain weight exhibited no significant difference with that of Xihui18 (Figure 3e; Appendix A).

#### 2.4.2. Analysis of Epistatic Effects of Two Non-allelic QTLs by DSSLs (D1–D2)

Pyramiding of *qGL9* (*a* = 0.23 mm) and *qGL12* (*a* = −0.19 mm) produced an epistatic effect of −0.23, which reduced the grain length of D2 by 0.19 mm in inheritance according to the genetic model of DSSL. The grain length of D2 (10.28 mm) was significantly shorter than that (10.66 mm) of Xihui18 and that (11.12 mm) of S3 with *qGL9*, while no significant difference was observed with that (10.28 mm) of S4 containing *qGL12*. The results indicated that *qGL12* was epistatic to *qGL9*. Pyramiding of *qGL5* (*a* = −0.08 mm) and a substitution segment without QTL for grain length on chromosome 7 yielded an epistatic effect of 0.20 mm, resulting in increasing the grain length of D1 genetically by 0.13 mm. The grain length of D1 (10.90 mm) was significantly longer than that (10.66 mm) of Xihui18, that (10.50 mm) of S1 containing *qGL5*, and that (10.65 mm) of S2 without QTL for grain length on the chromosome 7 segment (Figure 3b; Appendix A).

Pyramiding of *qGW5* (*a* = 0.64 mm) and *qGW7* (*a* = 0.21 mm) yielded an epistatic effect of −0.54 mm. Thus the genetic effect of D1 for grain width was 0.31 mm, according to the genetic model of DSSL. The grain width (3.63 mm) of D1 was significantly narrower than that (4.29 mm) of S1 with *qGW5*, but significantly broader than that (3.01 mm) of Xihui18 and that (3.43 mm) of S2 carrying *qGW7*. The results indicated that pyramiding of *qGW5* and *qGW7* with increasing effect of grain width yielded an intermediate grain width. Pyramiding of *qGW9* (*a* = 0.34 mm) and *qGW12* (*a* = 0.15 mm) produced an epistatic effect of −0.33 mm, which increased grain width of D2 genetically by 0.16 mm. The grain width (3.33 mm) of D2 was significantly narrower than that (3.70 mm) of S3 with *qGW9*, while significantly wider than that of (3.01 mm) of Xihui18, however, with no difference with that (3.31 mm) of S4 with *qGW12* (Figure 3c; Appendix A).

Pyramiding of *qRLW5* (*a* = −0.55) and *qRLW7* (*a* = −0.22) resulted in a positive epistatic effect of 0.50; according to the genetic model of DSSL, the genetic effect of D1 for the ratio of length to width was −0.27. The ratio of length to width (3.01) of D1 was significantly less than that (3.54) of Xihui18 and that (3.10) of S2 with *qRLW7*, while significantly greater than that (2.45) of S1 containing *qRLW5*. Pyramiding of *qRLW9* (*a* = −0.26) and *qRLW12* (*a* = −0.21) resulted in a positive epistatic effect of 0.26, which decreased the ratio of length to width of D2 by 0.23 genetically. The ratio of length to width (3.09) of D2 was significantly less than that (3.54) of Xihui18, while significantly greater than that (3.01) of S3 with *qRLW9*, and showed no significant difference with that (3.11) of S4 carrying *qRLW12* (Figure 3d; Appendix A).

Pyramiding of *qGWT5* (*a* = 4.65 g) and *qGWT7* (*a* = 0.35 g) resulted in a negative epistatic effect of −2.04 g, which increased the 1000-grain weight of D1 by 2.96 g. The 1000-grain weight (34.52 g) of D1 was significantly heavier than that (28.61 g) of Xihui18 and that (29.30 g) of S2 with *qGWT7*, whereas it was significantly lighter than that (37.91 g) of S1 carrying *qGWT5*. Pyramiding of *qGWT9* (*a* = 3.27 g) and a substitution segment without QTL for grain weight on chromosome 12 yielded an epistatic effect of −3.17 g, which genetically increased the 1000-grain weight of D2 by 0.10 g. The 1000-grain weight of D2 (28.98 g) was significantly lighter than that (35.14 g) of S3 with *qGWT9*, while no significant difference was observed with that (28.61 g) of Xihui18 and that (28.79 g) of S4 without QTL for grain weight (Figure 3e; Appendix A).

The above results indicated that different QTL pyramids can result in various epistatic effects, which is crucial to screen target QTLs in futural breeding by design.

## 3. Discussion

### 3.1. The Developed SSSLs and DSSLs Dissected from Rice Large-Grain CSSL-Z403 Are Valuable Pre-Breeding Tools

The core of plant breeding is the use of naturally occurring variation [24]. CSSL development can generate rich, naturally occurring variations. It is a valuable pre-breeding tool for expanding the genetic basis of existing cultivars, as well as excellent genetic material for QTL mapping and functional studies [5]. In this study, the large-grain rice CSSL-Z403 with 10 substitution segments was identified from progeny of the recipient parent Xihui18 and the donor parent Jinhui35 by advanced backcrossing combined with whole-genome SSR MAS. The CMS type of the Xihui18 being used as restorer comprised restoring genes *Rf1*, *Rf3* (Chr.1) and *Rf4* (Chr.10), and *Rf2* (Chr.2) and *Rf4* (Chr.10) in Jinhui35. The outstanding characteristic of Z403 was its large grain (40 g for 1000-grain weight) when compared with Xihui18. However, Z403 was still difficult to be used directly in precise rice breeding due to carrying too many substitution segments and QTLs. In order to create more ideal materials for direct breeding and basic study in gene functional analysis, four SSSLs and two DSSLs carrying QTLs for various grain sizes (*qGL5*, *qGW5*, *qGWT5*, *qGW7*, *qGWT7*, *qGL9*, *qGW9*, *qGWT9*, *qGL12* and *qGWT12*) were developed using F_2;3_ population from Xihui18/Z403. Intriguingly, the substitution intervals of these secondary substitution lines were not located in Xihui18 chromosome restoring genes (*Rf1*, *Rf3* and *Rf4*). Thus, they should serve as novel restorer lines. Moreover, single segment substitution line S3 carrying *qGL9* (a = 0.23 mm), *qGW9* (a = 0.34 mm) and *qGWT9* (a = 3.27 g) had larger grain. Its grain length, grain width and grain weight were 11.12 mm, 3.70 mm and 35.14 g, respectively, increasing significantly over those (10.66 mm, 3.01 mm and 28.61 g) of Xihui18. S1, S2 and S4 all had various grain sizes, which were larger than that of Xihui18. In addition, Liang et al. in our institute developed four secondary SSSLs harboring *qGL3-1*, *qGL3-2, qGL7, qGW3-2, qGW3-3, qGW7, qGW8, qGWT3-1*, *qGWT3-2* and *qGWT7* from donor Huhan3 in the genetic background of Xihui18 [6]. Sun et al. developed six SSSLs carrying *qGL1, qGL2, qGL3*, *qGL5, qGW1, qGW5-1*, *qGWT1, qGWT3* and *qGWT5-2* from donor Huhan3 in the genetic background of Xihui18, and S1, S3 and S5 increased yield per plant significantly [12]. Li et al. developed six SSSLs containing *qGL3*, *qGL5*, *qGL11*, *qGW5* and *qGWT5* from donor Huhan3 in the genetic background of Xihui18 [13]. These SSSLs carry a single substitution segment and specific identified QTLs from a donor parent in the genetic background of Xihui18. The multiple target QTLs located in different SSSLs can then be pyramided to construct target complex traits for breeding programs. Thus, they can be easily used in hybrid rice breeds by crossing with various sterile lines. In addition, these SSSLs can also be used in gene cloning and molecular mechanism analysis, and then conversely be used in precise breeding. It is worth mentioning that it has become a reality to pyramid favorable alleles for breeding new varieties by SSSLs as a platform. Using the library of 2360 SSSLs based on the rice AA genome of HJX74, Zhang et al. designed a series of novel varieties successfully, such as Huaxiaohei 1 (Guangdong Rice 2005015) and Huabiao 1 (Guangdong Rice 2009033). These achievements indicate that using SSSLs as a platform can facilitate designing and breeding varieties to meet production needs [23]. Therefore, these newly bred SSSLs of the excellent restorer line Xihui18 are valuable pre-breeding tools and promising materials applied to genetic basis research.

### 3.2. qGL6, qGL7, qGW7, qGWT7, and qGL9 Might Be QTLs Newly Identified by Comparison with Previously Reported Genes

In total, 16 QTLs were identified by both the secondary F_2_ segregating population of Xihui18/Z403 and the four developed SSSLs. *qGL5*, *qGW5*, *qRLW5*, and *qGWT5* were all linked to marker RM169 of the substitution segment on chromosome 5 of Z403, which might be involved in pleiotropy or close linkage. The genes *OsGSK2* [25] and *PPR5* [26] were in the substitution interval. The cell biology analysis suggested that *OsGSK2* regulates grain size by inhibiting cell expansion in the hull [25]. *PPR5* encodes a PPR protein that participates in mitochondrial function and endosperm development by controlling splicing of the intron 3 in the mitochondrial NADH dehydrogenase gene *NAD4*. In the rice endosperm mutant *ppr5*, the endosperm is chalky, shrunken, and white, and the starch granules are smaller, looser, and more spherical, resulting in a significant decrease in grain length, width, thickness, and 1000-grain weight [26]. Therefore, *OsGSK2* and *PPR5* might be potential candidate genes for *qGL5*, *qGW5*, *qRLW5*, and *qGWT5*. In addition, the QTLs were identified as *qGW5* and *qGWT-5-2* by Sun et al. [12], indicating that these QTLs should be major QTLs that can be inherited stably. However, they came from different donor parents, namely Jinhui35 and Huhan3. *qGW9*, *qRLW9*, and *qGWT9* were all linked to the same marker, RM3808, and might act as a pleiotropic QTLs or tight linkages. *OsSPL18* [27] was found in the substitution interval. Cytological analysis indicates that *OsSPL18* regulates glume development by affecting cell proliferation. It can bind to the *DEP1* promoter, suggesting that *OsSPL18* regulates spikelet development through the positive regulation of *DEP1* expression. Additionally, *OsSPL18* can be cleaved by *OsmiR156k*, involved in the *OsmiR156k*-*OsSPL18*-*DEP1* pathway of regulating rice grain number [27]. Therefore, *OsSPL18* could be a candidate gene for these QTLs. *qGL12*, *qGW12*, *qRLW12*, and *qGWT12* were all linked to the same marker RM491 of the substitution segment on chromosome 12 of Z403 and might be involved in pleiotropy or close linkage. Within this interval, *OsYUC11* is a key factor in auxin biosynthesis of rice endosperm, which affects grain filling and storage accumulation, leading to increasing starch accumulation and grain size [28]. Therefore, *OsYUC11* could be a candidate gene for *qGL12*, *qGW12*, *qRLW12*, and *qGWT12*. In addition, *qGL7*, *qGW7* and *qGWT7* (linkage to RM1135 at 16.88 Mb) in the study were different from the QTL with the same name (linkage to RM8261 at 23.92 Mb) detected by Liang et al. [6]. Thus, *qGL6*, *qGL7*, *qGW7*, *qGWT7*, and *qGL9* have not been reported and may be newly identified QTLs. These newly identified QTLs will provide a good basis for further functional analysis and breeding by design.

### 3.3. Identification of Epistatic Effects between Different Non-Allelic QTLs Is Essential for Breeding by Design

Most important agronomic traits such as grain size are controlled by many QTLs with minor genetic effects. Ando et al. found that the additive effect of each QTL was relatively small, and none of 38 QTLs for panicle architecture distributed on 11 chromosomes could explain much of the phenotypic difference in sink size between Sasanishiki and Habataki in 39 CSSLs [29]. Balakrishnan et al. argued that epistatic effects among QTLs are an important genetic component affecting gene pyramiding, especially for complex traits [19]. Epistasis is the interaction between two or more genes, and this interaction can be affected by allelic differences of each gene [30]. Disrupting co-evolving alleles can have unexpected consequences in plant breeding for complex traits, such as alleles no longer contributing to the trait of interest, or even affecting that trait in the opposite direction [31]. Thus, it is necessary to check whether non-allelic QTLs are involved in independent inheritance or epistatic interaction. Leng et al. detected five pairs of digenic epistatic interactions for rice grain size that yielded a highly phenotypic variation. They suggested that the epistatic effects should be considered during rice breeding [32]. The epistatic effects between QTLs for stigma exertion rate (*qSERb3-1*/*qSERb6-1*, *qSERb3-1*/*qSERb8-1*, and *qSERb3-1*/*qSERb12-1*) in three pyramiding lines were all positive and induced much larger pyramiding effects of 51.49%, 43.54%, and 55.51% [33]. In the study, using two DSSLs and four SSSLs dissected from progeny of Xihui18/Z403 as materials, we also found that pyramiding different QTLs for grain size produced various epistatic effects and yielded diverse grain sizes in rice. For example, pyramiding of *qGL9* (*a* = 0.23 mm) and *qGL12* (*a* = −0.19 mm) produced an epistatic effect of −0.23, resulting in a grain length (10.28 mm) of D2 that was significantly shorter than that (11.12 mm) of S3 with *qGL9*, while showing no significant difference with that (10.28 mm) of S4 containing *qGL12*, indicating that *qGL12* was epistatic to *qGL9*. Similarly, Li et al. also found that pyramiding *qGL3* (a = 0.43) and *qGL11* (a = −0.37) led to shorter grains in the dual-segment substitution line D2 and revealed that *qGL11* is epistatic to *qGL3* [13]. Liang et al. showed that pyramiding of short-grain QTLs (*qGL3–2* and *qGL3–1*; *qGL3–1* and *qGL7*) resulted in shorter grains [6]. Furthermore, in our study, pyramiding of *qGW5* (*a* = 0.64) and *qGW7* (*a* = 0.21) produced an epistatic effect of −0.54, resulting in increasing the grain width of D1 by 0.31 mm genetically. Finally, the grain width (3.63 mm) of D1 (carrying both *qGW5 and qGW7*) was intermediate of that (4.29 mm) of S1 with *qGW5* and S2 with *qGW7*. Liang et al. showed that pyramiding of broad-grain QTLs (*qGW3-2* and *qGW3-3*; *qGW3-2* and *qGW7*) resulted in broader grains [6]. All of these results indicated that different grain size QTL pyramids produce different epistatic effects and yield various grain sizes. Thus, it is very necessary to identify the epistatic effects for different non-allelic QTLs for grain size to realize breeding by design.

## 4. Materials and Methods

### 4.1. Experimental Materials

#### 4.1.1. Development of Z403

Z403 was developed using Xihui18 as the recipient parent and Jinhui35 as the donor parent. Xihui18 and Jinhui35 were all *indica* rice restorer lines bred by Southwest University, China. Xihui18 was a large panicle type rice restorer line with excellent comprehensive characters. Jinhui35 was a large-grain type rice restorer line. Firstly, 236 polymorphic markers were screened from the 429 simple sequence repeat (SSR) markers that covered the whole rice genome. Then, they were used to develop CSSLs from the BC_2_F_1_ to BC_3_F_7_. When the genotype of a marker was consistent with that of the recipient Xihui18, it was labeled as “A”, and when consistent with the donor Jinhui35, it was labeled as “B”. Heterozygous band types were labeled as “H”. The specific development progress was as shown in Figure 4. Finally, a novel large-grain CSSL Z403 with 10 substitution segments was identified in BC_3_F_7_. Chromosome substitution segments were identified as described by Liang et al. [6], and the estimated lengths of the chromosome substitution segments were calculated as described by Paterson et al. [34]. The chromosome map was constructed using the Mapchart 2.32 software developed by R.E. Voorrips, Plant Research International (https://www.wur.nl/en/show/Mapchart.htm, accessed on 22 December 2015).

#### 4.1.2. Materials for QTL Mapping

The QTL mapping population was derived from a secondary F_2_ segregating population consisting of 150 individuals raised from the cross between Xihui18 and Z403.

### 4.2. Experimental Methods

#### 4.2.1. Material Planting Methods

In June 2019, Xihui18 was crossed with Z403 to obtain hybrid seeds at the experimental station of Southwest University, Chongqing, China. Then, the hybrid seeds were planted at the Lingshui base in Hainan Province in September of the same year, and the F_1_ seeds were harvested. In March 2020, Xihui18, Z403, and the F_2_ population were sown at the experimental station of Southwest University. Then, 30 seedlings of each parental line and 150 individuals of F_2_ population were transplanted to the same experimental field in April, with 26.4 cm spacing between rows, 16.5 cm between hills, and 10 plants per row. In March and April 2021, Xihui18, Z403, and six individuals selected from the F_2_ population to develop SSSLs and DSSLs were sown and transplanted at the experimental station of Southwest University, using 30 plants per line. In March and April 2022, Xihui18, Z403, and four SSSLs and two DSSLs developed from the F_3_ population were sown and transplanted to the same experimental field, using 30 plants per line. Conventional management practices were applied.

#### 4.2.2. Assessment of Grain Size-Related Traits of All Materials

After maturation, 10 plants each of Xihui18 and Z403, four SSSLs and two DSSLs were harvested, together with 150 individuals of the F_2_ population. The total length and width of 10 grains lined up were measured with a 20 cm ruler with three replications for each plant and then used to calculate average per-grain values for each plant. The ratio of length to width was calculated as grain length divided by grain width. The 1000-grain weight of Xihui18 and Z403, as well as each substitution line, was measured from random samples of 3000 grains, where 1000-grain subsets were weighed on an electronic balance, with three repetitions. The 1000-grain weight of each F_2_ plant was determined as the weight of 200 grains, multiplied by 5, with three repetitions [6]. In addition, descriptive statistics, such as the mean values, standard deviation of these traits, and student’s *t*-test (two-tail) for comparison of these traits between 10 plants of Z403 and Nipponbare, were calculated using Microsoft Excel 2010 (15 June 2010).

#### 4.2.3. QTL Mapping

A total of 150 individuals of the secondary F_2_ population derived from Xihui18/Z403 were used for QTL mapping. DNA from each sample was extracted as the template of PCR using the cetyltrimethyl ammonium bromide (CTAB) method [35]. Fourteen SSR markers located in the substitution segments of Z403 were used as primer of PCR in QTL mapping. PCR amplification, polyacrylamide gel electrophoresis, and rapid silver staining were conducted following the methods described by Zhao et al. [36,37]. The Xihui18-type band was scored as “−1”, the Z403-type band as “1”, the heterozygote as “0”, and a missing band as “.”. The mean value for each trait from 150 F_2_ plants and the marker assignment value were used for QTL mapping. The mixed linear model (MLM) method implemented in the HPMIXED program of SAS 9.3 software (SAS Institute Inc., Cary, NC, USA) (http://suportsus.com/publishing (accessed on 21 June 2023), 2009, SAS/STAT: Users’ Guide, Version 9.3) was used to plot the QTLs. The significance level *p* < 0.05 was used as the threshold to determine whether a QTL was associated with the marker on the substitution segment [36].

#### 4.2.4. Development Method for Secondary SSSLs and DSSLs

Based on the QTL mapping results in 2020, 6 individual plants containing the target QTL and 0–2 heterozygous markers were selected from the F_2_ population and used to develop SSSLs and DSSLs by MAS. Each individual was planted as a line (Z854–Z859) in 2021. The leaves of 30 individuals per line were collected and used to extract DNA for genotyping by MAS using both the target substitution markers and residual heterozygous markers. The SSSLs and DSSLs were developed based on the rule that each substitution line contained only the homozygous target substitution segment, while the bands of the other markers were same as those of Xihui18 [13]. 

### 4.3. Identification and Analysis of Additive Effect of QTLs for Grain Size by 4 SSSLs

Each SSSL_i_ (S1–S4) was given the hypothesis (*H*_0_) that no QTL existed in the substitution segment of the SSSL_i_. When the *p*-value was less than 0.05 according to one-way analysis of variance (ANOVA) and LSD multiple comparison with Xihui18 by each SSSL_i_ in IBM SPSS Statistics 25.0, we rejected the hypothesis and considered that a QTL for a certain trait existed in the SSSL_i_. According to the genetic model in a certain environment (same year and same experimental field and no replicate plot designed), *P*_0_ = *μ*_0_ + *ε* for Xihui18 and *P_i_* = *μ*_0_ + *a_i_* + *ε* for an SSSL carrying a specific QTL, where *P*_0_ and *P_i_* represented the phenotypic value of any plant in a plot of Xihui18 and the SSSL_i_. *μ*_0_ represented the mean value of the Xihui18 population, *a_i_* represented the additive effect of the QTL from substitution segment of donor Jinhui35, whose positive effect indicated increasing phenotypic value and negative effect showed decreasing phenotypic value in substitution lines, and *ε* represented residual error [13]. Thus, the additive effect of the QTL was estimated as *a_i_ = (P_i_ − P*_0_*)/*2 (half of the phenotypic difference was estimated to be caused by inheritance). 

### 4.4. Analysis of Epistatic Effect between Two Non-Allelic QTLs for Grain Size in the DSSLs

For each DSSL_ij_, we assumed that *H*_0_: two loci (*Q*_1_ and *Q*_2_) controlling a certain trait located in substitution segments “*i*” and “*j*” were independently inherited, expressed as “2 + 0 = 1 + 1”. When the *p*-value was greater than 0.05 for *Q*_1_ × *Q*_2_ by two-way ANOVA analysis in IBM SPSS Statistics 25.0, we accepted the hypothesis that *Q_1_* and *Q_2_* in DSSL_ij_ were independently inherited. At this time, the phenotypic value of (Xihui18 + DSSL_ij_) was the same as that of (SSSL_i_ + SSSL_j_). By contrast, when the *p*-value was less than 0.05 for *Q*_1_ × *Q*_2_, we rejected the hypothesis and considered that an epistatic interaction had occurred between the two allelic loci *Q*_1_ and *Q*_2_, namely “2 + 0 ≠ 1 + 1”. According to the corresponding genetic model, Xiuhui18: *P*_0_
*= μ*_0_
*+ ε*, SSSL_i_: *P_i_ = μ*_0_
*+ a_i_ + ε*, SSSL_j_: *P_j_ = μ*_0_
*+ a_j_ + ε* and DSSL_ij_: *P_ij_ = μ*_0_
*+ a_i_ + a_j_+ I_ij_ + ε* (*μ_0_* was the average value of Xiuhui18, *a_i_* and *a_j_* were the additive effects of QTLs in substitution segments *i* and *j*, respectively, and their significance was determined by the corresponding *p* values of SSSL_i_ and SSSL_j_, *p* < 0.05 indicated the existence of *Q_1(2)_*, *I_ij_* was the epistatic effect between *Q*_1_ and *Q*_2_ in substitution segments “*i*” and “*j*”, and ε represented the residual error) [13]. The epistatic effect between *Q*_1_ and *Q*_2_ was calculated as *I = [(P*_0_
*+ P_ij_) − (P_i_ + P_j_)]/2* (half of the phenotypic difference was estimated to be caused by genetic factors). Finally, Duncan’s multiple comparison of grain size-related traits was performed on all SSSLs and DSSLs as well as Xihui18 using IBM SPSS Statistics 25.0.

## 5. Conclusions

A large number of SSSLs in the Xihui18 genetic background are developed from genetically diverse rice donors. In this study, a large-grain CSSL-Z403 was identified carrying 10 substitution segments (3.01 Mb average substitution length) from Jinhui35. In total, six QTLs for grain size were identified in five substitution segments on chromosomes 5, 6, 7, 9 and 12 of Z403, and they then were dissected into four novel SSSLs and two novel DSSLs. The large grain of Z403 was controlled at least by *qGWT5*, *qGWT7*, *qGWT9* and *qGWT12,* and was influenced through grain-length QTLs such as *qGL5*, *qGL6*, *qGL9* and *qGL12*, as well as grain-width QTLs such as *qGW5*, *qGW7*, *qGW9* and *qGW12*. Among 16 QTLs, four QTLs including *qGL6,* etc., might be novel compared with the reported documents. Again, by pyramiding analysis between two non-allelic QTLs such as *qGL9* and *qGL12*, *qGW5* and *qGW7*, *qGW9* and *qGW12*, *qGWT5* and *qGWT7,* etc., the results showed that different QTLs for grain size yielded various epistatic effects, which is essential for screening suitable QTLs in breeding by design. Consequently, these newly developed SSSLs and DSSLs harboring target QTLs in the genetic background of excellent restorer line Xihui18 will be valuable pre-breeding tools and promising materials applied to genetic basis research. The Xihui18-SSSL library will be a powerful platform for rice breeding by design in southwestern China.

## Figures and Tables

**Figure 1 ijms-24-12013-f001:**
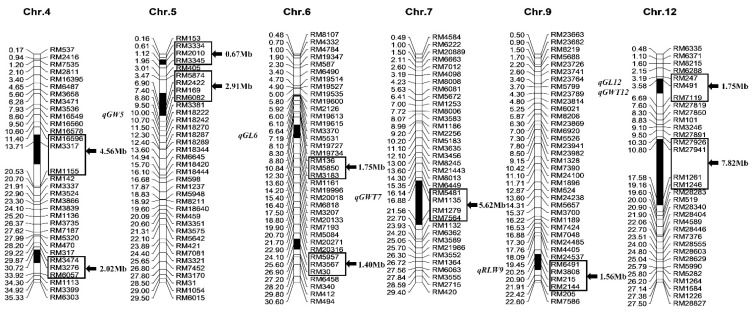
Substitution segments and detected QTLs of Z403 (the genome of *indica* rice “9311” was used as the reference). The physical distances (Mb) and QTLs detected are indicated on the left of each chromosome, the marker names and substitution segment length (markers in the box) are indicated on the right. GL, grain length; GW, grain width; RLW, ratio of length to width; GWT, 1000-grain weight.

**Figure 2 ijms-24-12013-f002:**
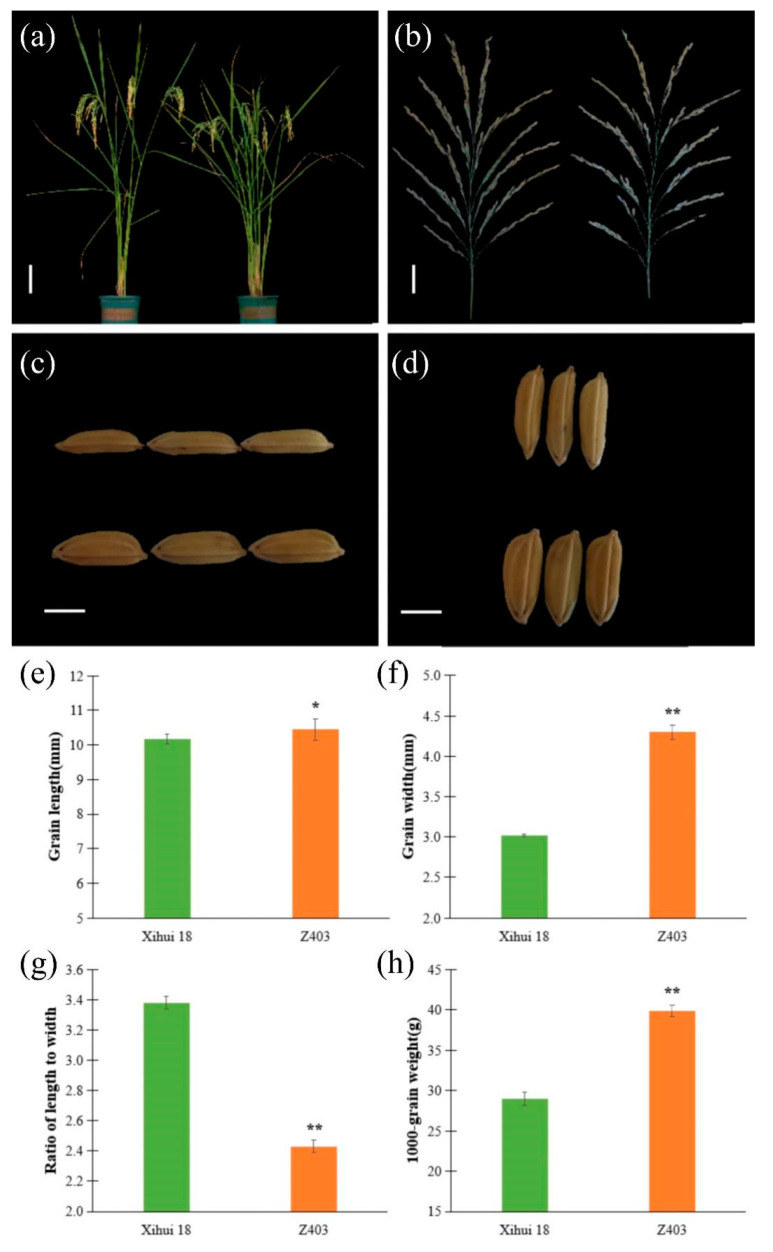
Plant type and grain size analysis of Xihui18 and Z403. (**a**) Plant type of Xihui18 (left) and Z403 (right). (**b**) Panicle of Xihui18 (left) and Z403 (right). (**c**) Grain length of Xihui18 (up) and Z403 (down). (**d**) Grain width of Xihui18 (up) and Z403 (down). Bars represent 20 cm in (**a**), 5 cm in (**b**), and 5 mm in (**c**) and (**d**). (**e**–**h**) Statistical analysis of the differences in four traits between Xihui18 and Z403. (**e**) Grain length, (**f**) grain width, (**g**) ratio of length to width, (**h**) 1000-grain weight. * and ** indicated significant differences at the 0.05 and 0.01 levels, respectively.

**Figure 3 ijms-24-12013-f003:**
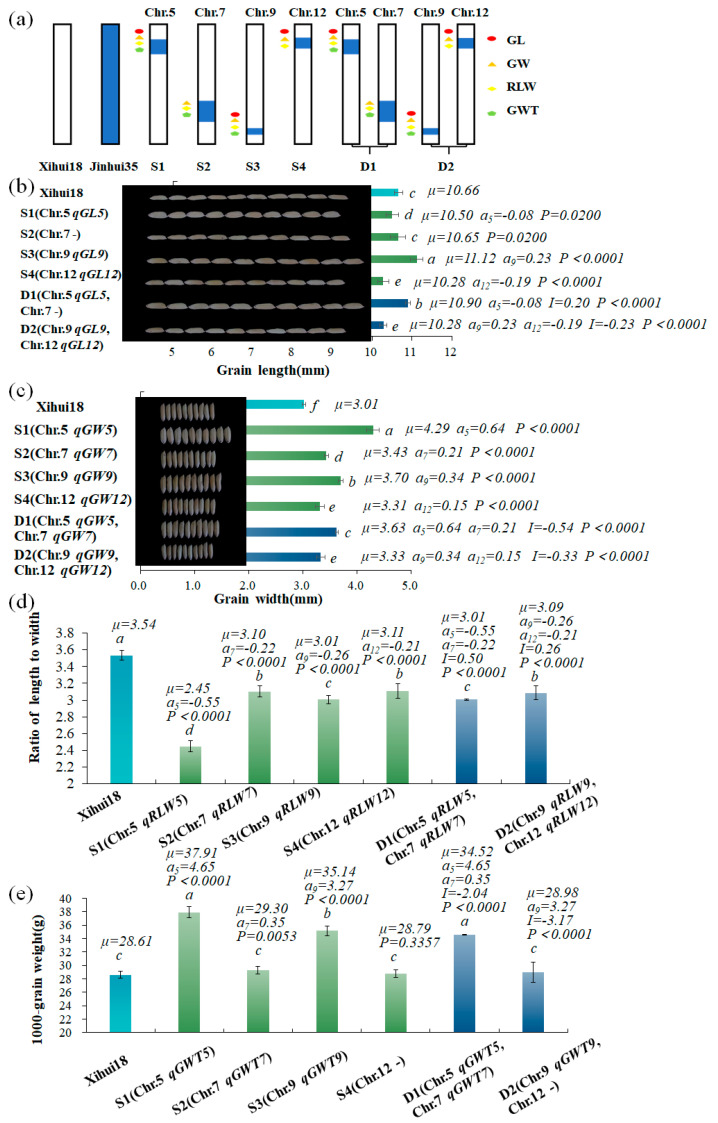
Additive and epistatic effects of QTLs for related traits in the SSSLs and DSSLs. (**a**) Diagram of the locations of substitution segments and QTLs in S1–S4 and D1 and D2. (**b**–**e**) Parameters of QTLs in different SSSLs and DSSLs, including grain length (**b**), grain width (**c**), ratio of length to width (**d**) and 1000-grain weight (**e**). Different lowercase letters on each top column indicate significant difference (*p* < 0.05), as determined by Duncan’s multiple comparison. *μ*: the average value of each line; *a_i_*: additive effect for each QTL controlling the trait, whose positive value showed allele from substitution segment increasing phenotypic value, while negative value decreasing one. *I:* epistatic effect between two allelic QTLs. *p* < 0.05 in SSSL indicated that a QTL existed in the substitution segment of the SSSL, as determined by one-way ANOVA and LSD multiple comparison with Xihui18; *p* < 0.05 in DSSL indicated that an epistatic effect of *Q1* × *Q2* existed in DSSL, as detected by two-way ANOVA. S1 (Chr.5: RM5874 (3.47 Mb)--RM2422 (6.90Mb)-RM169 (7.40 Mb)--RM6082 (8.80 Mb)); S2 (Chr.7: RM5481 (16.14 Mb)--RM1135 (16.88 Mb)-RM1279 (21.56Mb)--RM7564 (22.7 Mb)); S3 (Chr.9: RM6491 (19.45 Mb)--RM3080 (20.25 Mb)-RM215 (20.9 Mb)--RM2144 (21.91 Mb)); S4 (Chr.12: RM247 (3.19 Mb)--RM491(3.58 Mb)--RM7119(6.69 Mb)); D1 (Chr.5: RM5874 (3.47 Mb)--RM2422 (6.90Mb)-RM169 (7.40 Mb)--RM6082 (8.80 Mb); Chr.7: RM5481 (16.14 Mb)--RM1135 (16.88 Mb)-RM1279 (21.56Mb)--RM7564 (22.7 Mb)); D2 (Chr.9: RM6491 (19.45 Mb)--RM3080 (20.25 Mb)-RM215 (20.9 Mb)--RM2144 (21.91 Mb); Chr.12: RM247 (3.19 Mb)--RM491 (3.58 Mb)--RM7119 (6.69 Mb)). The internal markers connected with hyphens indicate the substitution segment from the donor, whereas the markers at each end of the substitution segment linked with “--” indicate that segment recombination might have occurred.

**Figure 4 ijms-24-12013-f004:**
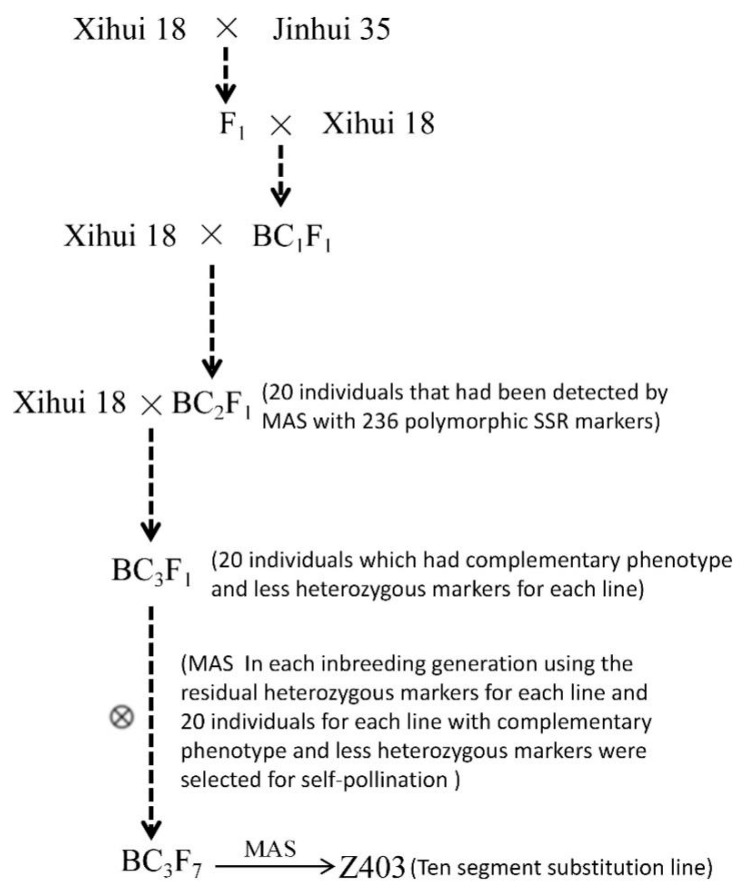
Strategy for the development of Z403. MAS: marker-assisted selection.

**Table 1 ijms-24-12013-t001:** QTLs for grain size identified in substitution segments of Z403.

Trait	QTL	Chr.	Linked Marker	Additive Effect	Var. (%)	*p*-Value
Grain length (mm)	*qGL6*	6	RM5850	0.08	8.54	0.0469
	*qGL12*	12	RM491	0.11	16.10	0.0137
Grain width (mm)	*qGW5*	5	RM169	0.09	19.15	0.0152
Ratio of length to width	*qRLW9*	9	RM3808	0.07	8.02	0.0395
1000-grain weight (g)	*qGWT7*	7	RM1135	3.99	19.49	0.0351
	*qGWT12*	12	RM491	2.76	9.34	0.0459

## Data Availability

Not applicable.

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
