# Peer review of "Genetic Effects Analysis of QTLs for Rice Grain Size Based on CSSL-Z403 and Its Dissected Single and Dual-Segment Substitution Lines"

_ijms, 2023, doi:10.3390/ijms241512013_

Round 1

Reviewer 1 Report

The presented work on genetic effects analysis of QTL in rice is highly significant for peer rice researchers in advancing knowledge about the genetic effects of QTLs on grain size and how these QTLs may help other genetics aspects (including epistasis effects) for continued breeding improvement. Overall, the presented study is very unique in nature and would certainly add new value to rice genetics research. The authors have presented their work succinctly and the manuscript can be considered to accept after major revision.

Please see the review comments attached herewith and revise the manuscript considering the made suggestions.

The authors have presented their work eloquently; however, there are some grammatical issues that need to address. I would recommend authors seek the help of a native English speaker or an English Editing Service to improve it further.  

Reviewer 2 Report

The research is devoted to the identification of QTLs controlling the grain size, an important character closely related to seed quality in rice. Therefore, the article fits the scope of the Special Issue “Molecular Research for Cereal Grain Quality 2.0”. The material investigated in the study, represents a large group of segment substitution lines, the derivatives of crosses with involving indica restorer line Xihui18 as recipient and various japonica lines as donors. Probably by this reason the experimental design and manuscript structure are very similar to those published in other paper devoted to the studying the analogous material (Sun, S., Wang, Z., Xiang, S. et al. Identification, pyramid, and candidate gene of QTL for yield-related traits based on rice CSSLs in indica Xihui18 background. Mol Breeding 42, 19 (2022). https://doi.org/10.1007/s11032-022-01284-x).

 In general, the data presented in the reviewed manuscript are completely original and are of special interest for the readers.

 Comments and recommendations:

1.       I recommend to extend the paragraph 2.1. Identification of Substitution Segments in Z403. The results of QTL mapping are described superficially. The reference to the source of reference genome is required.

2.       The corresponding section 4.2.3. QTL Mapping should be completed by the information on the source of SSR markers used in the study.

3.       Please give the information for which CMS type the Xihui18 and Jinhui35 being used as restorers and how this trait can be used in breeding programs with involving the newly developed SSSLs and DSSLs with target QTLs?

4.       The manuscript contains numerous grammatical errors and should be carefully edited.

Round 2

Reviewer 1 Report

The authors have made significant corrections based on the suggested review inputs and the revised manuscript looks much better than the earlier version.